# Adaptive Volt-Var Control Algorithm to Grid Strength and PV Inverter Characteristics

Toni Cantero Gubert [1] , Alba Colet [1,*] , Lluc Canals Casals [1,2,*] , Cristina Corchero [1,2] , José Luís Domínguez-García [1] , Amelia Alvarez de Sotomayor [3] , William Martin [4] , Yves Stauffer [4] and Pierre-Jean Alet [4]

1   Catalonia Institute for Energy Research (IREC), Sant Adrià de Besòs, 08930 Barcelona, Spain; tonicg8@gmail.com (T.C.G.); ccorchero@irec.cat (C.C.); jldominguez@irec.cat (J.L.D.-G.)
2   Department of Projects and Construction Engineering, Universitat Politècnica de Catalunya (UPC), 08034 Barcelona, Spain
3   Schneider Electric, 41092 Sevilla, Spain; amelia.alvarez@se.com
4   CSEM SA, 2002 Neuchâtel, Switzerland; william.martin@csem.ch (W.M.); Yves.STAUFFER@csem.ch (Y.S.); pierre-jean.alet@csem.ch (P.-J.A.)
*   Correspondence: acolet@irec.cat (A.C.S.); lluc.canals@upc.edu (L.C.C.)

**Abstract:** The high-penetration of Distributed Energy Resources (DER) in low voltage distribution grids, mainly photovoltaics (PV), might lead to overvoltage in the point of common coupling, thus, limiting the entrance of renewable sources to fulfill the requirements from the network operator. Volt-var is a common control function for DER power converters that is used to enhance the stability and reliability of the voltage in the distribution system. In this study, a centralized algorithm provides local volt-var control parameters to each PV inverter, which are based on the electrical grid characteristics. Because accurate information of grid characteristics is typically not available, the parametrization of the electrical grid is done using a local power meter data and a voltage sensitivity matrix. The algorithm has different optimization modes that take into account the minimization of voltage deviation and line current. To validate the effectiveness of the algorithm and its deployment in a real infrastructure, the solution has been tested in an experimental setup with PV emulators under laboratory conditions. The volt-var control algorithm successfully adapted its parameters based on grid topology and PV inverter characteristics, achieving a voltage reduction of up to 25% of the allowed voltage deviation.

**Keywords:** distributed power generation; low-voltage; test facilities; standards; voltage regulation; reactive power; microgrids; photovoltaic systems; optimization methods

## 1. Introduction

Distributed Energy Resources (DER), such as photovoltaic (PV) systems, are being increasingly integrated into distribution networks due to their low carbon emission when generating energy, an affordable price at small-scale level, and to the technology maturity as a strategy to face climate change [1].

Several problems appear when massive deployment of DER occurs, such as harmonics distortion, reverse power flows, or power losses [2]. Among them, overvoltage is the main potential problem at distribution level [3]. Currently, there are several strategies to correct voltage deviation and enhance grid stability, such as line refurbishing, on-load tap changers (OLTC), capacitor banks and static var compensators, battery storage, demand-side management and line voltage regulators, among others [4]. While most of the previous solutions are either expensive or difficult to integrate, the use of power electronics from PV inverters already installed is a more efficient and economical solution for network stability [5].

EN 50438 Standards and national grid codes allow for grid-tied PV inverters to participate actively in voltage regulation adjusting the exchange of reactive power [6]. However,

the compliance of standard EN 50160 [7], which indicates the maximum permissible voltage deviation at the Point of Common Coupling (PCC), is the one that fixes the maximum integration of DER. By absorbing or injecting reactive power, a smart inverter can correct over or under voltage deviations [8]. Voltage regulation is highly dependent on the grid topology where generation devices are placed [9], so a solution to increase the penetration of renewables has to consider the grid characteristics where DER are installed. The grid topology is defined by the resistance ($R$) and reactance ($X$) of power cables together with the short circuit ratio ($SCR$), which define the grid strength, i.e., the ability of the grid to maintain its voltage constant during the injection of active and reactive power from an energy source. The $SCR$ and $X/R$ ratio are the main indicators of grid strength [10].

Literature already presented the expression to quantify the overvoltage of a single DER in the distribution grid, taking into account grid characteristics and the amount of power that is generated by the PV inverter [11]. The voltage deviation from the PV inverter in the PCC ($U_{PCC}$) to the voltage source ($U_G$) would depend on the product of the grid resistance and active power ($P$) as a positive term in the equation and the product of the grid reactance and reactive power ($Q$) as a negative term. A volt-var control technique can be used to adjust the amount of reactive power based on the voltage level at PCC, so more reactive power is absorbed when the voltage deviation is bigger. Seuss et al. [12] proved its effectiveness as compared to other techniques, such as ramp-rate, fixed power factor, and volt-watt controls.

According to [13,14] and following IEC 61850-90-7 standard, the volt-var function can be managed by either autonomous DER units responding to local conditions or broadcasted from a centralized power system provider with the ability to understand the capacities of each DER. While a centralized control concentrates the processing capabilities in one high computational equipment, local control can work whenever there are communication problems with the centralized system [15]. In our study, the use of Remote Terminal Units (RTU) allows for both local and centralized control; acting as the gateway for each PV inverter and as the centralized system provider that pictures the whole grid. RTUs are commonly used to transmit data from electrical substations or remote areas to distributed control systems and, at the same time, they have processing capabilities to host algorithms to act locally.

An adaptive reactive power control is proposed in [16]; however, the control parameters are the same for all PV inverters to distribute the power demand, not being adapted to each unit. However, in our study, an RTU is installed close to the PCC obtaining local measurements and providing the parameters that define the volt-var control of each PV inverter; this versatile equipment gives a clear advantage in front of what has been exposed.

Nonetheless, volt-var has been used for other purposes, such as the minimization of line losses [17]. The present study not only evaluates this indicator, but it also extends its application with an algorithm design that is capable to adapting control parameters to the grid's topology. Its unique feature makes this algorithm valuable, since it is a global solution for both weak and strong grids. The effectiveness of updating the parameters of volt-var control has proved to be useful, yet only using simulations, even for different amount of PV penetration and weather conditions [18]. This variability is also considered in our study, where the algorithm provides optimized parameters that are based on the inverter's size to reduce voltage deviation.

While most of the studies only rely on simulation results [12,17,19], even at a grid scale [20,21], until recently [22] a step forward has been implemented in this work by doing experimental tests to prove the effectiveness of the algorithm and the control system for several five-day tests in a laboratory environment that emulates real conditions. A simplified system that counts on a grid emulator, a PV inverter, and an impedance emulating the grid's length are used in a microgrid laboratory to study the influence of a reactive power control algorithm in a power system without loads. This simplified representation of a power system in a two-bus equivalent model proved to be accurate when estimating the overvoltage impact due to PV across a distribution network [23], and it is the necessary step

to certify the functionality prior its testing at a larger scale pilot site in Greece in the framework of the H2020 SABINA project [24] (SmArt Bi-directional multi eNergy gAteway). There are multiple approaches to define and design the strategy of the algorithm in charge to parametrize the control system. For instance, artificial neural networks were proposed to understand the reaction of DER inverters [25]. However, the present study uses real power meter data to train the sensitivity matrix that describes the network behavior using heuristic optimization methods. Table 1 summarizes some of the works that have been done until now and their contribution, highlighting the fact that its implementation under laboratory conditions has yet not been performed for multiple grid environments.

**Table 1.** A summary of the literature review presented in the introduction.

| Reference | Year | Contribution |
|:---:|:---:|:---:|
| [9] | 2010 | Presents problems to connect PV in a grid |
| [3] | 2012 | Analysis of voltage profile problems due to DR penetration |
| [4] | 2016 | Review of control strategies for voltage regulation |
| [10] | 2016 | Presents the main indicators of the grid topology |
| [11] | 2016 | Expression to quantify the overvoltage of a single DER |
| [12] | 2016 | Simulates the efficiency of volt-var control techniques |
| [17] | 2016 | Voltage regulation can be used to minimize line-losses |
| [20] | 2016 | Simulations of Volt-var regulation at grid scale |
| [5] | 2017 | Indicates how to do testing in laboratories. No implementation |
| [8] | 2017 | Indicates the possibility of inverters to regulate voltage |
| [18] | 2017 | Analyzes the benefits of adapting the parameters |
| [13] | 2018 | Presents centralized management of volt-var control techniques |
| [15] | 2018 | Theoretical and simulation analysis of local volt-var control |
| [23] | 2018 | Simplified power system in a two-bus equivalent model |
| [14] | 2019 | Presents centralized management of volt-var control techniques |
| [16] | 2019 | Adaptive reactive power control for PV for ultra-weak grids |
| [19] | 2019 | Analysis of the behavior of grids with high PV deployment using volt-var control chain Strategies |
| [21] | 2019 | Simulations of Volt-var regulation at grid scale |
| [25] | 2019 | Use of Artificial Neural Networks for Volt-var control |
| [22] | 2020 | Hundreds of loads and generation simulations to evaluate the impact of control methods |

The main contributions of this paper are:

- The presentation of a new algorithm that adapts the volt-var parameters to different network conditions (strong or weak grids) and PV inverter characteristics based on power meter data at the DER level to reduce overvoltage and line loading. Moreover, the algorithm provides different solutions that are based on the PV penetration and power factor levels of the distribution grid.
- The validation of its effectiveness in a laboratory with real equipment and with communication elements, such as RTUs, being a bridge between simulation environments (which is the most common case in the literature) and a large-scale deployment (which will be done further in the SABINA project).

## 2. Materials and Methodology

### 2.1. Adaptive Volt-Var Control Algorithm

The volt-var control function is characterised by five parameters that are represented in the curve shown in Figure 1. Parameters $Q_{max}$ and $Q_{min}$ represent the inverter's reactive power capabilities; $u_{min}$ and $u_{max}$ determine the dead-band where the reactive power is not exchanged, and $d$ is the droop parameter that corresponds to the slope of the curve. With $u_{meas}$ being the voltage that is measured at the PCC expressed in per unit (p.u.) and $S_{nom}$

the total apparent power of the inverter, the resulting reactive power can be computed, as follows, Equations (1) and (2)):

$$Q_{inj} = 100 \cdot (u_{min} - u_{meas}) \cdot S_{nom}/d; \quad Q_{abs} = 100 \cdot (u_{max} - u_{meas}) \cdot S_{nom}/d \quad (1)$$

$$Q_{max} \leq Q_{inj} <= Q_{min}; \quad Q_{max} \leq Q_{abs} <= Q_{min} \quad (2)$$

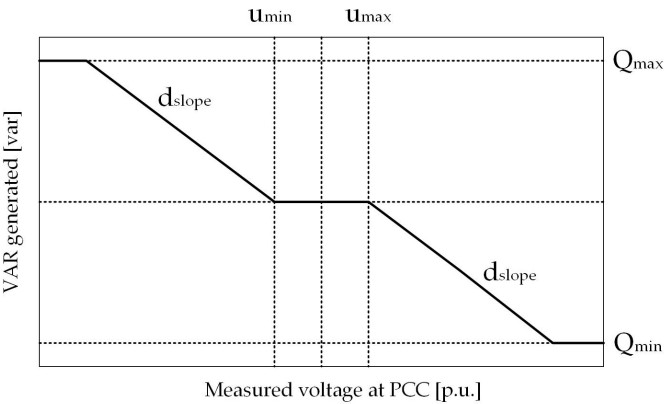

**Figure 1.** $Q(u)$ curve of volt-var control: reactive power absorbed or injected as a function of the measured voltage at the PCC.

The current practice in Europe consists in setting these parameters to fixed values when the PV system is installed. The values are defined in grid connection rules. Many distribution system operators follow the German application rule VDE-AR-N 4105 and mandate the following values $Q_{min} = -Q_{max}, Q_{max} \simeq 0.44 S_{nom}, u_{min} = 0.97, u_{max} = 1.03$, $d \simeq 9.18$. To adapt to the specific environment of each PV system, a new algorithm has been developed in the framework of the SABINA project, which finds out the optimal parameters of volt-var control based on power meter data [26]. The main difficulties when determining these parameters is the need of a whole simulation of the electrical grid, with detailed information regarding network data, PV systems location, and characteristics. These data are usually not available or accessible. The aim of the algorithm is to determine volt-var control optimal parameters without the need of a simulation program to represent the electrical grid. For this reason, a voltage sensitivity matrix is used to represent an approximation of the grid's topology avoiding the usage of precise network details. The voltage sensitivity matrix is derived from the power flow in Equations (3) and (4), where $P_i$, $Q_i$, and $U_i$ are the measured active and reactive power and the voltage at node $i$, respectively, $G_{ij}$ is the real part of the admittance of the line connecting node $i$ to node $j$, $B_{ij}$ is the imaginary part of the admittance, and $\theta_i$ is the voltage phase of node $i$.

$$P_i = U_i \sum_{j=1}^{n} U_j(G_{ij}cos(\theta_i - \theta_j) + B_{ij}sin(\theta_i - \theta_j)), \quad (3)$$

$$Q_i = U_i \sum_{j=1}^{n} U_j(G_{ij}sin(\theta_i - \theta_j) + B_{ij}cos(\theta_i - \theta_j)) \quad (4)$$

Voltage sensitivity matrix is a known method that is described in [14,27,28] that allows for predicting the change in voltage if measurements of power are available, which can be easily obtained with basic metering equipment.

The implementation strategy of the algorithm follows four steps:

1.  Training period: the sensitivity matrix is trained setting non-optimal control parameters and monitoring voltage, active and reactive power. The volt-var parameters will vary randomly every minute in a range of pre-defined values, trying to repre-

sent an average standard parametrization. The training periodn lasts two days with measurements every minute, having a total of 2880 points.

2.  Reference scenario: the volt-var control is turned off and the voltage, active and reactive power are measured to obtain a reference scenario. These values will be compared with the ones obtained in the scenario with optimized parameters. The reference scenario period lasts five days with measurements every minute, having a total of 7200 points. Even though the active and reactive power are known parameters that are imposed to the PV emulator, as it is explained in Section 4, a mismatch exists between the imposed values and the real ones.

3.  Offline optimization: the otpimization is done off-line and it follows two steps. First, a black-box model is used to define an initial feasible set of volt-var parameters. Second, the Nelder-mead method is used to obtain the optimal volt-var parameters, where the cost function is a trade-off between minimizing the voltage deviation and the line current increase. In [26], the black-box model and heuristic implementation are detailed and compared with further methodologies, this analysis is out of the scope of this paper.

4.  Evaluation period: the parameters that are found by the algorithm in step 3 are implemented to the PV inverter. Again, the period of analysis lasts five days with measurements every minute, having a total of 7200 points. These results are compared with the reference scenario ones.

Figure 2 illustrated the implementation strategy of the algorithm with the four steps together with other relevant data for the experiment definition.

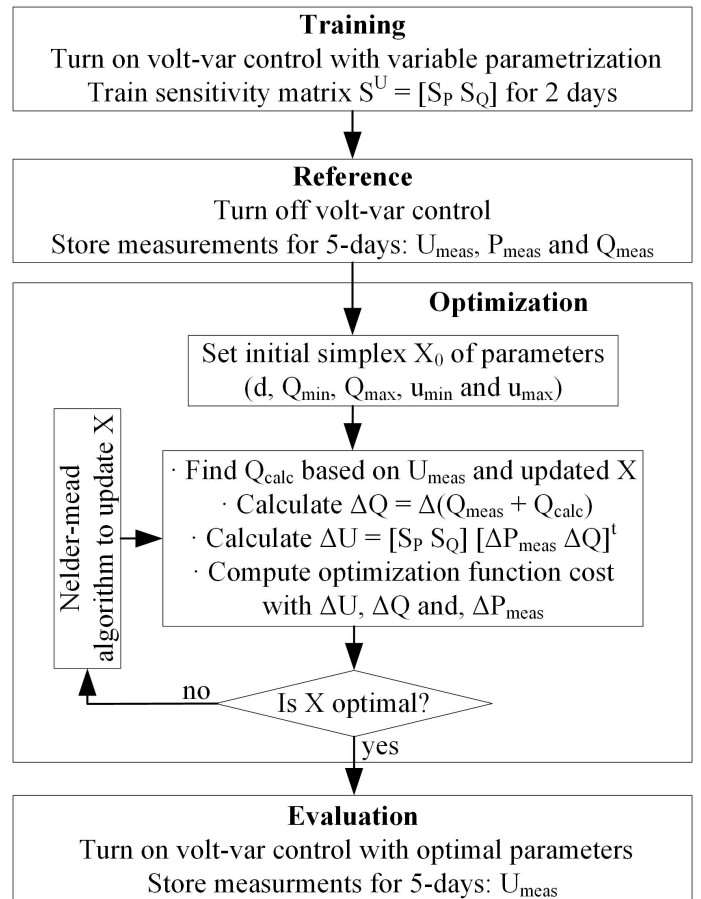

**Figure 2.** Implementation strategy flowchart.

### 2.2. Two-Bus Equivalent Model

Experimental validation of the volt-var algorithm with actual PV inverters is a shortcoming from previous studies that allows for testing its effective performance, with its behaviour being connected with real communication devices (such as RTUs), and serving as a previous step for deployment at a larger-scale pilot in the same SABINA project, as mentioned in the introduction. Subsequennly, to test the effectiveness of the reactive power control algorithm in a physical and controlled environment, some simplifications have to be done in terms of LV network representation. A simplified model is used to accurately estimate the magnitude of overvoltage within LV areas with limited data in a small amount of time. Such data are summarized while taking into consideration the impedance of each branch and the parent branch to which they are connected, the load/generator ratings, and the parent connection. This simplification allows representing the LV area with a two-bus model formed by three components: a slack bus with a defined reference voltage, an equivalent network impedance, and a PQ bus for the PV generation. Bus $B_0$ is the slack bus, representing the connection of the grid at a fixed reference voltage $U_G$, as shown in Figure 3. Bus $B_1$ connects the PV generation ($S_{PV}$) to the grid through an equivalent impedance ($Z_{eq}$). $U_{PCC}$ is the maximum voltage in the LV area during tests, which represents the furthest point from the voltage source.

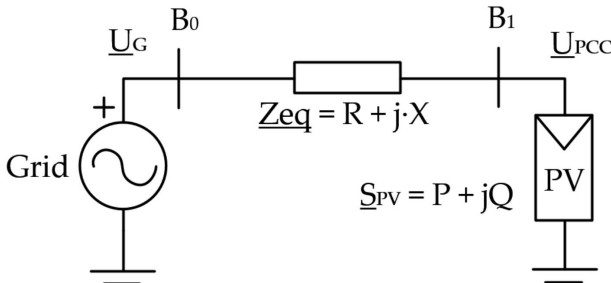

**Figure 3.** Two-bus equivalent model of a full network after reduction.

From the previous representation and assuming a small variation in the angle between $U_G$ and $U_{PCC}$, Equation (5) is obtained:

$$\Delta \vec{U} = \vec{U_{PCC}} - \vec{U_G} \approx (R \cdot P + X \cdot Q)/U_{PCC} \tag{5}$$

The two-bus model is very useful for preparing the tests in the microgrid laboratory and linking the grid topology with the size and location of the inverter. It presents the relation of the total equivalent resistance ($R$) and reactance ($X$) as a function of the nominal voltage at which the reference voltage is imposed ($U_G$), the total apparent power of the inverter ($S_{PV}$), and the common indicators of grid characteristics ($SCR$ and $X/R$ ratio). The development of Equations (8) and (9) comes from the definition of the $SCR$ (Equation (6)), as the ratio between the short circuit capacity ($SCC$) of the grid against the rated power of the energy source ($S_{PV}$). Equation (7) relates the equivalent impedance with $X/R$ ratio ($XRR$):

$$SCR = SCC/S_{PV} = U_G/(|Z_{eq}| \cdot S_{PV}]) \tag{6}$$

$$|Z_{eq}| = \sqrt{R^2 + X^2} = R\sqrt{(1 + XRR^2)} = X\sqrt{(1 + XRR^{-2})} \tag{7}$$

$$R = (U_N^2 \cdot \sqrt{1 + XRR^2})/(SCR \cdot S_{PV}) \tag{8}$$

$$X = (U_N^2 \cdot \sqrt{1 + XRR^{-2}})/(SCR \cdot S_{PV}) \tag{9}$$

These equations are used to define the resistance and reactance values of the test design.

*2.3. Energy Smart Laboratory*

The facilities of the Catalonian Institute for Energy Research (IREC) count with the Energy Smart Laboratory (SmartLab) have a configurable AC three-phase network, which interconnects several power electronics converters, battery storage systems, and power load banks. SmartLab is based on a hardware emulation approach that allows physical equipment to operate under a broad range of scenarios that represent real conditions without depending on the boundary conditions of specific equipment and, thus, being suitable for experimental validations [29,30]. For the specific case of this experiment, testing uses two main groups of elements in SmartLab:

1. The power electronics system, in Figure 4, which counts on:

   - A grid emulator that acts as voltage source setting up the reference voltage and frequency of the emulated microgrid. For the cases in this study, all of the tests are fixed at 400 V and 50 Hz.
   - An emulated distribution line that has variable inductance and resistance in the range from 0 to 20 Ohms for both elements.
   - A PV emulator with 4-kVA maximum apparent power.

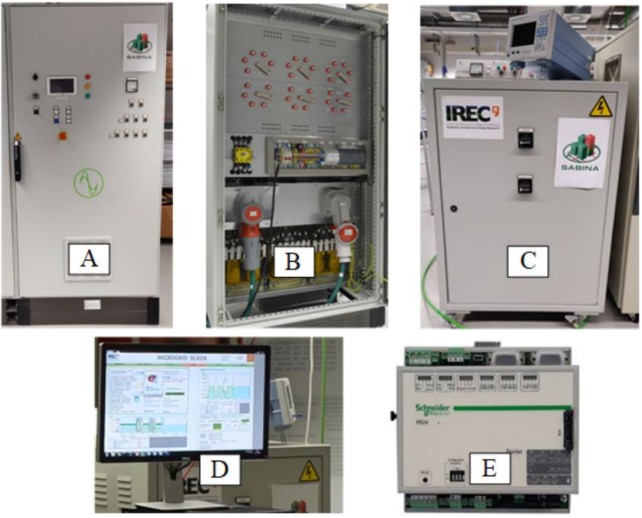

**Figure 4.** Key components of the experimental setup. (**A**) Grid emulator, (**B**) Line emulator, (**C**) Emulated photovoltaic (PV), (**D**) Supervisory Control and Data Acquisition (SCADA), and (**E**) Remote Terminal Units (RTU).

2. The control and communications system, in Figure 5, where we can find:

   - A PV emulator adapted to meet the requirements for the study. This emulator calculates the reactive power to absorb based on the parameters of the volt-var curve. The inverter also collects electrical measurements and then sends them to the RTU.
   - The Schneider Electric RTU [31] provides the parameters from the volt-var curve once they are calculated depending on the control mode that was selected by the adaptive algorithm. The RTU has the possibility to host the algorithm locally or to use it remotely, in which case the RTU communicates through MQTT to an external server with processing capabilities. A patent in this regard is being published by Schneider Electric.
   - A Supervisory Control and Data Acquisition (SCADA) controls the equipment in the experiment and it gathers the monitored information. The communication between the RTU and external server is also tracked.

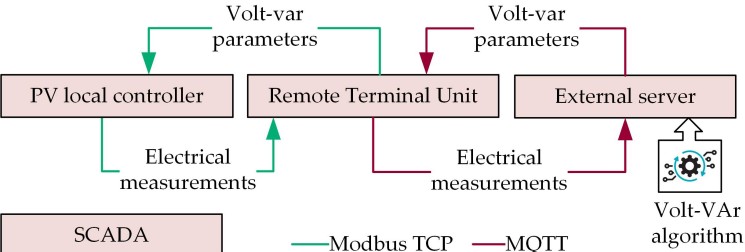

**Figure 5.** Communication interaction between components. The color of the arrow expresses the protocols used. The algorithm is hosted in an external server.

## 3. Case Study

### 3.1. Test Design

Because of the considerations and capabilities of the microgrid in SmartLab, tests are designed to evaluate the effectiveness of the reactive power control under different grid topologies and optimization modes. The study presents the results for two scenarios:

- Scenario 1 (S1): refers to a strong grid with short distribution line, this is a grid topology with large $SCR$ and $X/R$ ratio. It represents the scenario in which PV inverters are installed near an electric substation, so it is less sensitive to voltage changes.
- Scenario 2 (S2): refers to a weak grid with a large distribution line, this is a grid topology with small $SCR$ and $X/R$ ratio. It represents the scenario in which PV inverters are installed in remote areas that are more sensitive to voltage changes.

An $X/R$ ratio of 0.5 is considered to be low enough to represent the weak grid topology of S2, being twice more resistive than inductive. High values of resistance will produce a voltage rise at PCC that should not exceed the EN 50160 regulation; to avoid PV emulator malfunctioning, the maximum resistance in the setup is calculated to not overpass the 10% of nominal voltage in case of maximum power injection. Subsequently, from Equation (5), when considering $S_{PV} = 4$ kW as the maximum rated power of the PV inverter, a nominal voltage of red $U_G = 400$ V and neglecting the effect of reactive power, Q = 0 var $P = 4000$ W, the maximum resistance to set in SmartLab is 4.4 $\Omega$. Taking $R = 4\,\Omega$ to avoid reaching the limits, the remaining parameters $SCR$ and $X$ are obtained.

For the strong grid topology of S1, an $X/R$ ratio of at least 3 is taken to have an inductive-driven grid. When considering that 0.7 $\Omega$ is the minimum resistance that can be set in the laboratory, the values for $X$ and $SCR$ are obtained. The resulting parameters are exposed in Table 2:

**Table 2.** Grid topology parameters defined for each scenario.

|  | *SCR* **Ratio** | *X/R* **Ratio** | $R\,[\Omega]$ | $X\,[\Omega]$ |
|---|---|---|---|---|
| S1 | 18.1 | 3 | 0.7 | 2.1 |
| S2 | 9.0 | 0.5 | 4.0 | 2.0 |

A reactance of 2 $\Omega$ is close to the real value for a low voltage distribution grid (0.6 to 3.5 $\Omega$). Those values include the impedance of the transformer and a length of 300 m in the case of S1 and of 1600 m for S2.

Figure 6 shows the active power that is injected by the PV emulator. This pattern is obtained from real PV panels and it is scaled so the maximum power injected is 4000 W, as this is the limit of PV emulator power in SmartLab. As indicated, the same five-day PV production pattern is followed both in reference and evaluation periods and for all scenarios. The real data used to train the sensitivity matrix during the training period correspond to the first two-days of the same pattern.

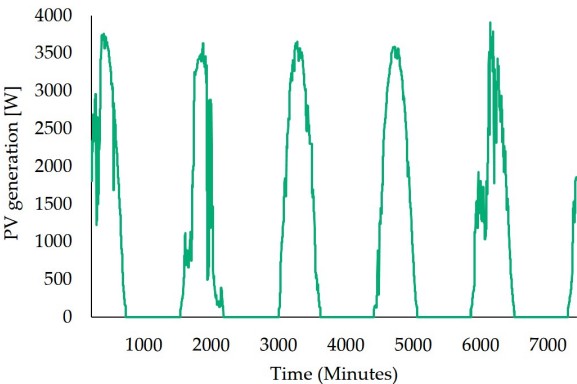

**Figure 6.** PV generation for a five-day test. The starting point corresponds to 9 a.m.

As mentioned in Section 2.1, every test follows four steps: training period, reference period, optimization, and evaluation (Figure 2). This steps are repeated not only for the different scenarios, but also for the different optimization modes tested.

Because voltage regulation with reactive power is a trade-off between correcting voltage deviation and the increase of line current, three different optimization modes of balancing these indicators are tested: The mixed, the full voltage, and the balanced modes.

- The mixed (M) mode gives the same weight to the cost function to minimize voltage deviation and line current increase.
- Thefull voltage (FV) mode aims to achieve the maximum voltage reduction compared to any of the other control modes.
- The balanced (B) mode falls between the mixed and the full voltage mode giving more importance to the voltage deviation reduction in the cost function as compared with the mixed mode.

Thus, the testing phase is composed of six tests summarized in Table 3: three tests with S1 and three tests with S2 evaluating all different optimization modes. The reference and the training periods are done only once per scenario, while the optimization and evaluation steps need to be conducted in all six tests.

**Table 3.** Summary of the scenarios, periods, and optimization modes that were carried out in each test.

| Test | T1 | T2 | T3 | T4 | T5 | T6 |
|---|---|---|---|---|---|---|
| Scenario | 1 | 2 | 1 | 2 | 1 | 2 |
| Training | x | x | | | | |
| Reference | x | x | | | | |
| Optimization | x | x | x | x | x | x |
| Evaluation | x | x | x | x | x | x |
| Optimization mode | M | M | FV | FV | B | B |

### 3.2. Key Performance Indicators (KPI)

To test the effectiveness of the volt-var parameters that were found by the adaptive algorithm, the reference and evaluation periods are compared for different scenarios and optimization modes. Because the purpose of this study focuses on overvoltage situations, the parameters that are analyzed in Section 4 are $U_{max}$, $Q_{min}$ and droop. Nonetheless, the following KPIs are also analyzed:

- Droop effectiveness (volt/volt): indicates how many Volts the reactive power control can reduce compared with the reference period where there is no voltage regulation. The maximum value possible is 1, i.e., for each volt increased at the PCC, the voltage control can reduce it to 1 volt. The minimum value would be 0, which corresponds to the reference case where no volt-var control is applied. The number is obtained when

considering all the points that fulfill the following two conditions: the line voltage is higher than $U_{max}$ and the reactive power less than $Q_{min}$.

- 95th percentile voltage reductio (%): points out the voltage reduction that a specific volt-var control is able to achieve. To better see its effectiveness, this value is expressed in relation to the maximum voltage deviation allowed by the standards (10% of the nominal voltage), $\Delta U_{max} = 40$ V. The 95th percentile value is taken instead of the maximum voltage value to avoid singular points (i.e., when there is no reactive power availability, the same maximum voltage values are obtained in both the reference and in the evaluation periods). This indicator points out that for 95% of the test, voltage values at the PCC are kept below that number, complying with the EN50160 standard.
- Line current increase (%): this indicator calculates the average line current during 5 days when the voltage is higher than $U_{max}$, comparing the reference with the evaluation periods.
- Power factor: is the ratio between the total active power and total apparent power supplied by the inverter. When reactive power control is enabled, the total apparent power increases and, therefore, the power factor reduces. Again, this indicator is calculated as the average for five days when the voltage is higher than $U_{max}$.

## 4. Results and Discussion

In this section, the tests described in Section 3.1 are implemented and the obtained results are summarized. Table 4 lists the optimized parameters regarding overvoltage for the defined tests (Table 3): droop, $U_{max}$ and $Q_{min}$. A lower droop indicates a higher slope for the volt-var curve, i.e., for each volt increase at the PCC more reactive power will be absorbed by the PV inverter.

**Table 4.** Summary of the optimized parameters for the six evaluation periods.

| Test | Scenario | Optimization Mode | Droop | $Q_{max}$ [pu] | $Q_{min}$ [var] |
|------|----------|-------------------|-------|----------------|-----------------|
| T1 | 1 | M | 5.789 | 1.001 | −794 |
| T2 | 2 | M | 35.69 | 1.001 | −2114 |
| T3 | 1 | FV | 1.000 | 1.005 | −1694 |
| T4 | 2 | FV | 7.748 | 1.005 | −4000 |
| T5 | 1 | B | 1.001 | 1.010 | −2999 |
| T6 | 2 | B | 8.030 | 1.010 | −1364 |

In Figure 7, the effectiveness of the droop parameter is plotted by the scenario and optimization mode. Lower values of droop mean that the volt-var control is more effective. However, there are other parameters that should be considered, such as grid topology. Section 4 discusses the limitations for low values of droop parameters in weak grid topologies. Additionally, note that, although not presented in Figure 7, droop effectiveness has also been evaluated during the learning period in order to compare the results against a generic voltage control system. For this two-day lapse, the values of the droop indicator were of 0.39 and 0.09 for S1 and S2, respectively. These results show, effectively, better voltage performance than without any control system; nonetheless, they are less effective than all three modes analyzed in this study.

Taking into account that the effectiveness of the control relies on the capacity to reduce voltage at the PV inverter location, Figure 8 shows the 95th percentile of the maximum voltage reduction that was reached at the PCC in relation to the maximum allowed deviation.

Figure 8 shows how, for a weak topology grid, like the one in S2, the adapted control parameters is capable of reducing the voltage up to 25% in comparison to the maximum voltage deviation (10 V), even though droop parameters in S2 were larger than in S1.

To have a clearer picture of the voltage reduction capacity of the algorithm and how results improve after the two days learning period, Figure 9 plots the monitored voltage of the five-day test for the reference and the evaluation period in S2 using the FV optimization

mode. The maximum voltage reduction of 25% is retrieved from the 95th percentile straight lines (green for the reference case and red for the evaluation period).

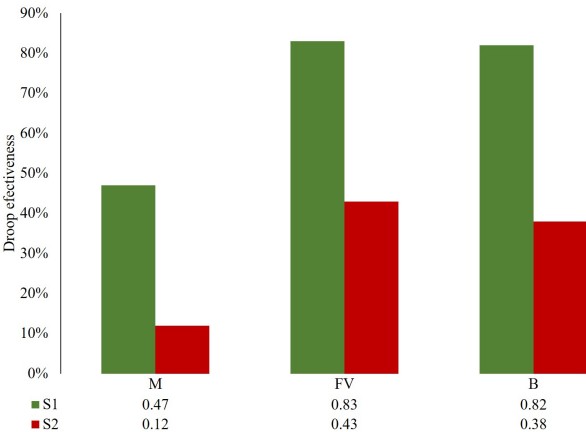

**Figure 7.** Droop effectiveness from evaluation periods of all tests.

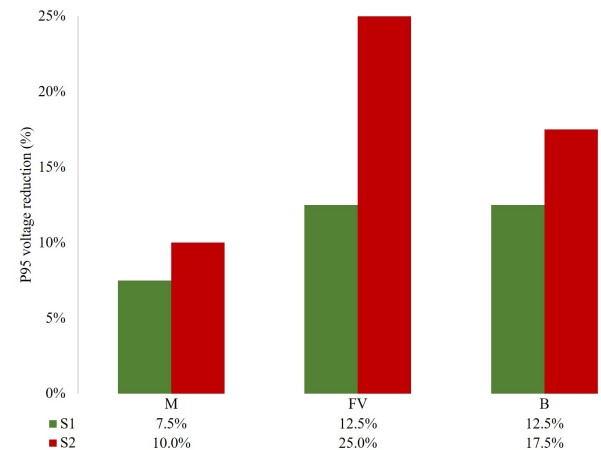

**Figure 8.** 95th percentile voltage reduction relative to maximum allowed deviation.

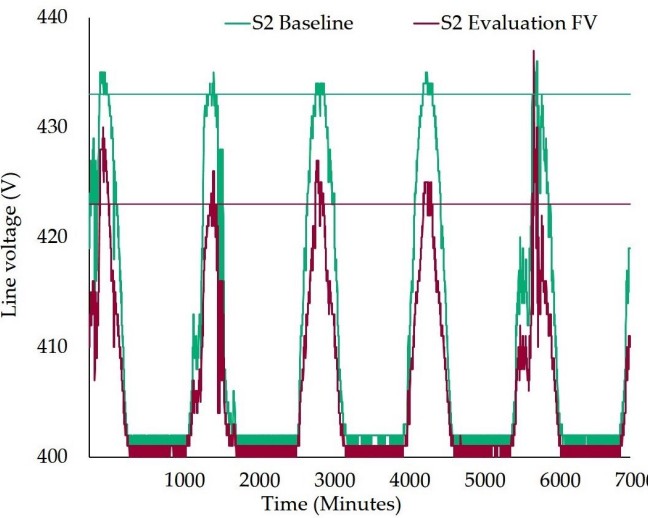

**Figure 9.** Line voltage for 5-day test. Straight lines indicate the 95th percentile from the maximum voltage.

The PV inverter threshold for the reactive power ($Q_{min}$) is another factor that affects the effectiveness of the control system. Lower thresholds of $Q_{min}$ allow more reactive power

absorption and, consequently, more current flowing through the lines. The current increase that results from the different scenarios under the three voltage regulation control modes is visible in Figure 10, presenting similar trends to those that were observed in the voltage reduction. That is, the best results are observable using the FV mode in S2, although the current increase in S1 is also close to the 25% for both the FV and B modes. In any case, the M mode presents the less performing response for this indicator (current increase), similarly to what was observable with the droop parameter or the voltage reduction.

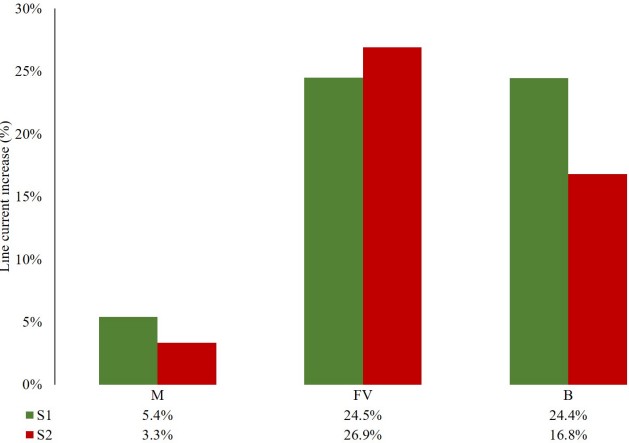

|  | M | FV | B |
|---|---|---|---|
| ■ S1 | 5.4% | 24.5% | 24.4% |
| ■ S2 | 3.3% | 26.9% | 16.8% |

**Figure 10.** Line current increase (%) from evaluation periods as compared with reference periods for all tests.

Nonetheless, line current increase is also conditioned by the droop effectiveness. For that reason, the M optimization mode presents a higher line current increase for a strong grid topology than for a weak one, even though the threshold is higher ($Q_{min}$ = −794 var in S1 and $Q_{min}$ = −2113 var in S2).

However, it is important to mention that lower values of $Q_{min}$ lead to lower values of power factor, which, in some cases, might reach undesired values for the proper operation of the grid. Figure 11 presents the power factor values observed in all tests. Notice how the configuration that presented better results in the previous indicators (S2 using a FV mode) achieves the worse results with regards to the power factor. Values of around 0.7 are hardly acceptable in most cases and, thus, the M mode for any grid topology is better placed.

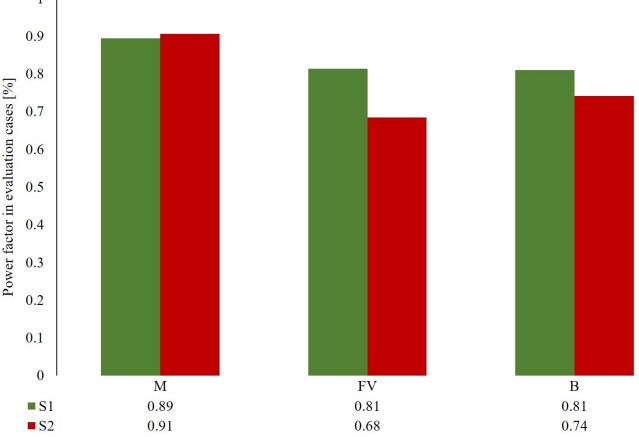

|  | M | FV | B |
|---|---|---|---|
| ■ S1 | 0.89 | 0.81 | 0.81 |
| ■ S2 | 0.91 | 0.68 | 0.74 |

**Figure 11.** Average power factor for all five-day evaluation periods when voltage at the PCC is above $Q_{max}$.

### 4.1. Optimized Volt-Var Parameters

Volt-var control algorithm has been tested in different scenarios showing the strengths and weakness of each optimization mode for each case. The M mode results to be the

less efficient in terms of voltage reduction, but the most conservative in terms of current increase. On the other hand, the FV mode is the one achieving higher voltage reduction in both scenarios, while the B mode gives an intermediate solution between the other two modes. Nevertheless, the results obtained are discussed in this section, giving some recommendations on when each mode is more suitable to be used beyond voltage deviation and line current reduction.

A small droop parameter together with a low threshold of $Q_{min}$ and $Q_{max}$ close to nominal voltage seems to be the best option to achieve higher voltage reduction. However, tests show that small values of droop also entail reaching the reactive power limit sooner. Whenever this threshold has a low value ($Q_{min}$ is negative), the PV emulator works at low power factor values most of the time. In this study, the power factor of the PV inverter has not been limited, but the fact that active power is prioritized in front of reactive power, when there is high PV generation, there is less reactive power availability, because the active power reaches the nominal power of the PV inverter. In those cases, the voltage increases both for the unavailability of reactive power and for the increase of active power. This effect is represented in Figure 12, where the dashed green line represents the designed behaviour that was previously presented in Figure 1.

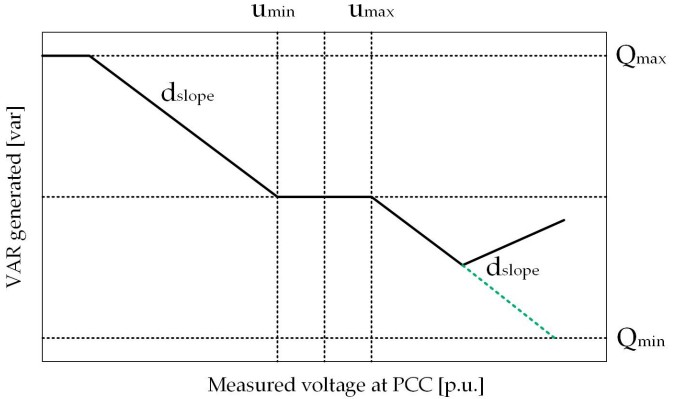

**Figure 12.** Effect on reactive power unavailability in volt-var curve.

The preferred optimization mode to avoid this effect is the one that better fits the grid requirements: For weak grids with large PV penetration, the use of higher values of $Q_{min}$ ($Q_{min}$ is negative) is preferred, so there is enough reactive power availability when there is more active power generation. For instance, reactive power limit in test 6 (S2, B) $Q_{min} = 1364.974$ var is bigger than in test 4 (S2, FV) $Q_{min} = 3999.747$ var. Subsequently, in test number 6, less voltage reduction is achieved, but there has been reactive power availability for almost all the five-day test, especially in moments of large active power generation when voltage increase is particularly high. Thus, test 6 is preferred than test 4 in terms of grid stability and, in those cases, the use of the B optimization mode is suggested to be the most appropriate. In the situation of both strong grids and weak grids with low penetration of PV and low line loads, the preferred optimization is be the FV, because more voltage reduction is provided. In some cases, though, when loads' power factor is not inductive, the use of FV mode is not advised, since the power factor descends below the recommended value.

Finally, the M optimization mode is recommended in highly loaded lines, especially in weak grids, when it is more important to minimize the current increase to the power lines, but still reduce the overvoltage.

In summary, the effect of the grid strength matters when looking for a volt-var control that is able to adapt to the grid topology where the inverter is installed. Large values of reactance entail a voltage reduction while large values of resistance mean less voltage reduction. In the cases studied, the maximum voltage reduction is achieved in Test 4 using

the FV optimization mode for S2, with almost 25% (10 V) when compared to the maximum voltage deviation (40 V).

### 4.2. The Added Value of Going from Simulations to Real Testing

Working with real equipment in a microgrid under laboratory conditions lead to some issues that needed interventions that are not present in analytical calculations or simulated environments. Over the multiple situations encountered, the following list presents the most relevant challenges that were faced during the testing phase:

- Unbalanced grid voltage: grid emulator's power electronics uses the direct-quadrature-zero transformation to simplify its control. This transformation should be used in balanced systems, otherwise, the control will lead to undesired operation of the converter. However, the test was performed having different equipment in each line: metering equipment, PV emulator and distribution line impedance. Because of the unbalanced consumption of those elements, the system was not balanced, so the control parameters in the grid emulator were adjusted to compensate the voltage grid unbalances. It is important to take this fact into account when doing the following step towards the larger-scale pilot site in SABINA and on any other grid environment, as the totally balanced situation is rather unrealistic.
- Oscillating reactive power setpoint: voltage measurements from the internal voltmeter in the PV emulator provide integer values to the control algorithm. This fact caused discontinuities on the reactive power setpoints, leading, in some cases, to an oscillating and inaccurate control. An increase in the sampling time of the voltmeter solved such inaccuracy for laboratory conditions. Nonetheless, larger-scale installations do tend to have even less precise equipment, which should be considered when adapting the algorithm.
- Inaccurate closed-loop control: internal closed-loop control of active and reactive power of PV emulator was not accurate in the whole range of power. Even though the sensors are calibrated and the control adjusted, still there is up to 3% inaccuracy between setpoint and sensor measures. This is expected to occur, even with worse precision, when deploying these solutions at a larger scale in real stablished grids.

### 4.3. Additional Power Quality Considerations

Note that, for all of the tests carried out, the line current is very small when compared to the maximum line capacity, because the test has been designed for a system with only one power generation unit. In fact, cable capacity is between 100 and 200 A and the maximum current provided by the PV emulator is around 5 A. Despite that line loading is not an issue for the specific tests executed in this study, the increase of current when using reactive power control has been proved.

Power factor is the consequence of the active and reactive power that the control algorithm sets to the emulator, in some tests a value as low as 0.68 (Figure 11) is obtained, on average, for a five-day test. This exaggeratedly low values are not acceptable in terms of distribution grid recommendation. However, distribution lines are inductive and most of the loads too, then a capacitive reactive power behaviour of the emulator as the ones obtained in this study can compensate the typical inductive performance of electric grids.

Finally, THD must be less than 5% for general application according to EN50600 standard. During all of the tests, the values remained within the regulation limits, and no correlation is found between the increase of THD and reactive power use of the PV emulator. To demonstrate how far from the limit the tests were, Figure 13 plots the THD and reactive power measured at each minute of the evaluation period of Test 4.

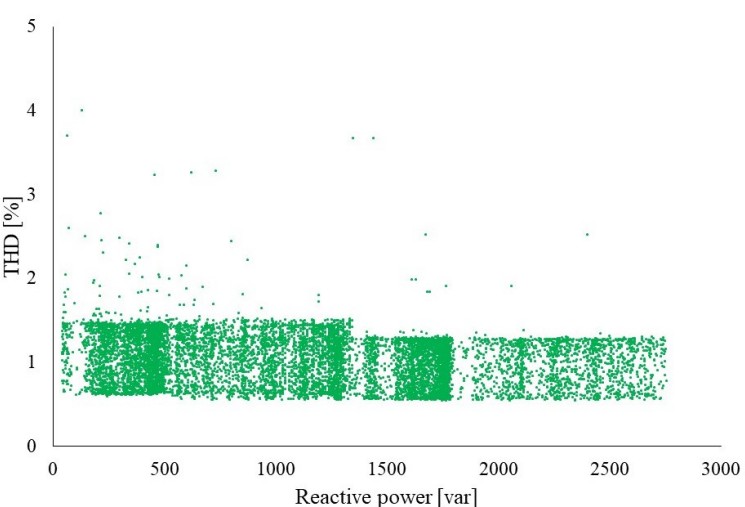

**Figure 13.** Correlation between reactive power and THD is almost 0.

## 5. Conclusions

This paper has showb a novel adaptive volt-var control algorithm being presented and validated. This study has demonstrated the performance and efficacy of applying such algorithms in PV inverters for voltage regulation in a laboratory environment. The research has proven that the control must adapt to the network where the PV inverter is connected and the current grid status (e.g., line loading, PV penetration, weak or strong grid, etc.) by modifying its control parameters to ensure optimal response and regulation.

In addition, three different methodologies have been presented, depending on the optimization objective to fulfill, since there is a trade-off between the voltage deviation and current increase. In this sense, the B optimization mode better fits weak grids with large PV penetration, while the FV would be preferred in both strong grids and weak grids with low penetration of PV and low line loadings. However, the M optimization mode would be more useful in weak grids with highly loaded lines.

Relevant outcomes of this study indicate that a low value of droop parameter and reactive power thresholds are not always desired when looking for a large voltage regulation and fulfill the system operation requirements, including standards, grid codes, and power quality restrictions. Moreover, reactive power availability in PV inverters is relevant and a capacitive power factor presents an advantage for inductive-driven grids.

Moreover, the integration of RTU with PV systems in a microgrid is shown to be beneficial for data acquisition, which can be used for both algorithm training and operation. The usage of both RTU and PV power meter data improve the response of the algorithms due to enhanced knowledge of the grid status, and it also allows for a local or remote control strategy that might solve eventual communication losses in real scale applications. Among the studied scenarios, a voltage reduction of up to 25% is achieved while keeping THD within the regulation limits. Such outcomes, together with the possibility to use RTUs with a local or a centralized control, appeals to the network operator to enhance the grid stability for different grid topologies.

**Author Contributions:** Conceptualization, W.M., Y.S. and P.-J.A.; Data curation, T.C.G.; Formal analysis, W.M. and Y.S.; Funding acquisition, C.C. and Y.S.; Methodology, T.C.G., A.C., L.C.C. and A.A.d.S.; Supervision, L.C.C., C.C., J.L.D.-G. and P.-J.A.; Validation, W.M. and Y.S.; Writing—original draft, T.C.G.; Writing—review & editing, T.C.G., A.C., L.C.C., C.C., J.L.D.-G., Y.S. and P.-J.A. All authors have read and agreed to the published version of the manuscript.

**Funding:** The European Community Horizon 2020 program under project SmArt BI-directional multi eNergy gAteway (SABINA), grant agreement 731211, has financially supported this research.

**Institutional Review Board Statement:** Not applicable.

**Informed Consent Statement:** Not applicable.

**Data Availability Statement:** Refer to contacts in SABINA website or ask the authors.

**Conflicts of Interest:** The authors declare no conflict of interest.

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
