# Peer review of "Adaptive Volt-Var Control Algorithm to Grid Strength and PV Inverter Characteristics"

_sustainability, doi:10.3390/su13084459_

Round 1

Reviewer 1 Report

This study presents a centralized algorithm for providing local Volt-VAr control parameters to each PV inverter, based on the electrical grid characteristics where each equipment is installed. Since accurate information of grid characteristics is typically not available, the parametrization of the electrical grid is done using power meter data in DER location and a voltage sensitivity matrix. The algorithm has different optimization modes to both minimize voltage deviation and line current. The paper is well-analysed, however, I have the following concerns.

  1. Writing quality should be improved using a professional service
  2. The contribution of the work should be further highlighted after finding the research gap
  3. Recent literature review with critical analysis should be presented in the Abstract. I would suggest using a table to summarise the literature.
  4. It is better not to put an equation in the abstract for an explanation of the problem. It can be done in the problem formulation section.
  5. More simulation and experimental work need to be conducted.
  6. The quality of the figure should be improved.
  7. A comparative analysis needs to be conducted.

Author Response

The authors want to thank the reviewer for the contributions and comments, which certainly will improve the quality and understandability of the paper. 

This study presents a centralized algorithm for providing local Volt-VAr control parameters to each PV inverter, based on the electrical grid characteristics where each equipment is installed. Since accurate information of grid characteristics is typically not available, the parametrization of the electrical grid is done using power meter data in DER location and a voltage sensitivity matrix. The algorithm has different optimization modes to both minimize voltage deviation and line current. The paper is well-analysed, however, I have the following concerns.

The authors want to thank the reviewer for the positive feedback and proceed to answer the comments done.

1. Writing quality should be improved using a professional service

The article has been thoroughly reviewed according to the reviewer notice. The reviewer will notice that there is "red" sentences, marking the changes, all over the manuscript.

2. The contribution of the work should be further highlighted after finding the research gap

The authors want to thank the reviewer for this note. In this new version, the contribution is highlighted at the end of the introduction, during the discussion and in the conclusions section. Some clarifications were added all over the text. Most important remarks are related to the algorithm being not only adaptative to electric grid conditions, like the line impedance and length (grid strength), but also that it proves to be a solution to the increase in PV penetration and to its related power factor level in the distribution grid. In addition, it is worth noting that the validation of such algorithm in a laboratory environment with real equipment has allowed to gain new insights and potential risks or limitations that the real-field may pose on such developments. 

3. Recent literature review with critical analysis should be presented in the Abstract. I would suggest using a table to summarise the literature.

The authors do agree with the reviewer that the analysis in the introduction should be critical. Up to seven new references were added in this section and over the manuscritp. The text has changed considerably to both clarify and highlight the contribution of our research. Nonetheless, after having re-written the text, the authors consider that to include a table summarising the literature review used in the introduction would be somehow repetitive.  

Note that the authors understood that the reviewer was referring to the introduction section and not the abstract, as it is not possible to do what the reviewer mentions in the abstract according to the journal’s instructions. 

4. It is better not to put an equation in the abstract for an explanation of the problem. It can be done in the problem formulation section.

Thank you for this point. As required, the equation has been moved to the section “Two-bus model” where it is referenced. 

5. More simulation and experimental work need to be conducted.

The authors do agree with the reviewer that more tests would be desirable to further justify the results obtained. However, we do regret to inform that, as part of a research project, tests ended up in summer 2019 and the equipment is now dismantled and working for other tests.  

Additionally, it is impossible to do these tests in the 10 days revision time given by the journal, as tests last between 5 and 7 days (preparation and learning time) and further results would have to be analysed. 

For these reasons, doing more tests is impossible for having nor the time neither the facilities to do so at the moment.

6. The quality of the figure should be improved.

As suggested by the reviewer the authors verified the quality of all images. Now all of should be in really good quality (at least we verified to see them in good conditions, we hope that they do not change during the reviewing process).    

7. A comparative analysis needs to be conducted.

The authors consider that the study already presents a comparison of three modes of control in contrast to not having any. However, in order to assess the request from the reviewer, another comparison has been added for the droop parameter (figure 7) based on the generic control system used in the “learning period”. For this 2-days period, the values for the parameters in the control system were variable so the algorithm could choose the best ratio according to the balanced, mixed or full voltage premisses. An example for the droop effectiveness index is presented in the new version of the manuscript. During this period, the response of the system was worse using this control than using the chosen ones. In fact, the values of the droop test are of 0.39 and 0.09 for scenarios S1 and S2 when they are higher in all the other cases (see figure 7 in the manuscript) showing the voltage regulation potential. 

Reviewer 2 Report

In this manuscript, a Volt-var control algorithm is proposed and validated in lab. The matter is of interest and the design of the research method looks adequate. Before it can be accepted, some corrections and improvements are needed though.

First, the English spelling and grammar should be reviewed by a native speaker. Here are a list of mistakes that should be corrected and things that could be improved or reviewed.

- We can read VAr many times in the manuscript, but the symbol of reactive energy was officially adopted as var, kvar or Mvar, lowercase, without capital letters. Please review it in title, abstract, text (lines 26, 50, 55, 68, 70, and so on), and even figures (1, 4, 6, and so on) or table 3.

- There are other mistakes repeated along the text, like the use of “consist on” instead of “consist of” in lines 80, 110, 124, 129, 136, 141, and so on. Other mistakes or expressions to review or improve could be the following ones. The use of “to both minimize” does not look very clear in line 10. Standards “allow” instead of “allows” in line 31, The use of “in front” is not clear in line 52. Space missing in line 89. Scenario instead of “Scenaio” in table 3. The use of “and its effectiveness proven” in line 320 is unclear. Preferred instead of “preferred” in line 403. The regulation specifies instead of “specify” in line 408. In general, please review the text and, if possible, ask a native speaker to correct all mistakes conveniently to improve the quality.

As regards the manuscript, it lacks some essential analysis and information that make the work impossible to be reproduced. The presented work should be explained so that other researchers can apply it and can understand the method, the advantages, the limitations and so on. In the introduction, a thorough analysis of other works with similar purposes should be included. In the materials and methods, all steps should be presented carefully or at least cited. For example, what does Nelder-mead algorithm consist of? Are there other works that use it? Please, review sections 1 and 2 to include all the necessary information to be able to follow exactly the steps of the experiments developed.

Finally, I know that it is not possible to test the presented method with a real scale example, and the simulation is fine as a proof of concept, but the authors should have considered to include more and longer simulation periods to better validate the method. Despite that, I would like to thank the authors for including the simulation as an extra step to validate the proposal.

After including the commented improvements, I suggest to improve the conclusions and try to highlight the advantages or main contributions of this proposal.

In my opinion, these comments should be considered before acceptance. I think that these comments could guide the authors and help them increase the quality of the paper. I suggest including them or explaining why they are not considered.

Thank you in advance for considering my comments.

Author Response

The authors want to thank the reviewer for the contributions and comments, which certainly will improve the quality and understandability of the paper. 

In this manuscript, a Volt-var control algorithm is proposed and validated in lab. The matter is of interest and the design of the research method looks adequate. Before it can be accepted, some corrections and improvements are needed though.

The authors want to thank the reviewer for the feedback given and proceed to answer all the comments done. 

First, the English spelling and grammar should be reviewed by a native speaker. Here are a list of mistakes that should be corrected and things that could be improved or reviewed.

- We can read VAr many times in the manuscript, but the symbol of reactive energy was officially adopted as var, kvar or Mvar, lowercase, without capital letters. Please review it in title, abstract, text (lines 26, 50, 55, 68, 70, and so on), and even figures (1, 4, 6, and so on) or table 3.

- There are other mistakes repeated along the text, like the use of “consist on” instead of “consist of” in lines 80, 110, 124, 129, 136, 141, and so on. Other mistakes or expressions to review or improve could be the following ones. The use of “to both minimize” does not look very clear in line 10. Standards “allow” instead of “allows” in line 31, The use of “in front” is not clear in line 52. Space missing in line 89. Scenario instead of “Scenaio” in table 3. The use of “and its effectiveness proven” in line 320 is unclear. Preferred instead of “preferred” in line 403. The regulation specifies instead of “specify” in line 408. In general, please review the text and, if possible, ask a native speaker to correct all mistakes conveniently to improve the quality.

As suggested by the reviewer, the text has been carefully revised and changed to improve its English following the list of mistakes indicated above and many other wordening identified. The reviewer will see that the new version of the manuscript is marked in red (showing the changes) all over the text. The authors hope that this new version satisfies the english quality standards of the reviewer.

The authors want to thank the reviewer for noticing notation and units wrongly written. Changed to var. Changed Volt to volt according IEC https://www.iec.ch/si-units 

As regards the manuscript, it lacks some essential analysis and information that make the work impossible to be reproduced. The presented work should be explained so that other researchers can apply it and can understand the method, the advantages, the limitations and so on. In the introduction, a thorough analysis of other works with similar purposes should be included.

As suggested by the reviewer, up to seven new references were included in the introduction and all over the manuscript to present other works tackling similar goals. 

In the materials and methods, all steps should be presented carefully or at least cited. For example, what does Nelder-mead algorithm consist of? Are there other works that use it? Please, review sections 1 and 2 to include all the necessary information to be able to follow exactly the steps of the experiments developed.

The authors want to thank the reviewer for highlighting this issue. The algorithm is a heuristic method better explained in reference [24], which has been added in the text for those that want to enter into more detailsThis information I now readable in sections 2.1 and 2.2, which have been thoroughly revised to respond to the reviewer’s request

Finally, I know that it is not possible to test the presented method with a real scale example and the simulation is fine as a proof of concept, but the authors should have considered to include more and longer simulation periods to better validate the method. Despite that, I would like to thank the authors for including the simulation as an extra step to validate the proposal.

The authors do agree with the reviewer that more tests would be desirable to further justify the results obtained. However, we do regret to inform that, as part of a research project, tests ended up in summer 2019 and the equipment is now dismantled and the facility is working for other projects.  

Additionally, it is impossible to enlarge these tests in the 10 days revision time given by the journal, as tests last between 5 and 7 days (preparation and learning time) and further results would have to be analysed. 

For these reasons, doing more tests (not simulations) is impossible for having nor the time neither the facilities to do so at the moment.  

Nonetheless, the method is being validated at larger-scale deployment in the pilot site in Greece. When results are available, the authors would be eager to share them in an expansion of the present work. 

After including the commented improvements, I suggest to improve the conclusions and try to highlight the advantages or main contributions of this proposal.

As suggested by the reviewer, the conclusions section incorporates some of the advances and main contributions of the work done and presented in this study. The discussion section has been also changed so it presents other relevant findings. Thank you again for noticing this point. 

In my opinion, these comments should be considered before acceptance. I think that these comments could guide the authors and help them increase the quality of the paper. I suggest including them or explaining why they are not considered.

Thank you in advance for considering my comments.

The authors are the ones thanking the revision done by the reviewer. Thank you for these interesting comments that, effectively, will improve the quality of the paper in the cases where they were applicable.  

The authors hope to have adressed the comments so the reviewer is satisfied with the new version of the manuscript.

Round 2

Reviewer 1 Report

The paper has improved the quality of the manuscript, however, I do have the following concerns:

  1. The authors have overlooked the following comment. It is important to improve readability to include a summarised table in the Introduction. I will encourage you to do that. "Recent literature review with critical analysis should be presented in the Abstract. I would suggest using a table to summarise the literature"
  2. Small paragraphes does not look good while reading a manuscript. Please correct it through the manuscript.
  3. The discussion Section is misleading. I guess this should belong to the Result Section with different scenarios. Please correct it.
  4. It is not a valid excuse to execute the comments of "5. More simulation and experimental work need to be conducted." We should focus on genuine research work NOT any dummy research work.

Author Response

First of all, the authors want to thank the reviewer for his time and effort in doing the revisions.

Here the reviewer will find the responses to the points raised

  1. The authors have overlooked the following comment. It is important to improve readability to include a summarised table in the Introduction. I will encourage you to do that. "Recent literature review with critical analysis should be presented in the Abstract. I would suggest using a table to summarise the literature"

Dear reviewer, we regret to read that we overlooked your comment. That was certainly not the intention. In our previous answer we clearly indicated that it gave us the impression that, adding the summary table would make the paper maybe a little repetitive but that, if the reviewer considered it necessary, we would do it. Thus, as suggested another time by the reviewer, the table summarizing the literature is now readable at the end of the introduction with the addition of two more recent articles in the field.

  1. Small paragraphes does not look good while reading a manuscript. Please correct it through the manuscript.

As suggested by the reviewer, the manuscript has, now, longer paragraphs whenever it was possible to do so.

  1. The discussion Section is misleading. I guess this should belong to the Result Section with different scenarios. Please correct it.

As suggested by the reviewer, the section of “discussion” has been removed.

  1. It is not a valid excuse to execute the comments of "5. More simulation and experimental work need to be conducted." We should focus on genuine research work NOT any dummy research work.

The authors are shocked to read that the reviewer considers the present work a “dummy research work”. The work presented here is the result of the first stage of a H2020 research project. These results were accepted and validated by the European commission. This study here presents what was done in the laboratory using commercial RTUs to control the voltage regulation with an adaptive algorithm that can work either in a local or centralized mode. These RTUs are now being installed in a real scale application, as a second phase of the project, in the pilot site of the project in Greece. Whenever results are available, be certain that the authors will analyze them and proceed to enlarge the work done in the present manuscript.

Reviewer 2 Report

Dear authors. Thank you for considering my comments. I think that the paper looks much better now. I only suggest to improve figure titles if possible. In the new version, some figures say VAr instead of var (figures 1, 2, 5, 12 and 13 contain var with wrong usage of upper/lower-case). I recommend to review this.

After these changes, I have no more objections.

Thank you for your work.

Author Response

Thank you for your effort and sharp eye. We were so focussed doing all the changes in the text in the previous submission that the fact that figures had volt-var written incorrectly passed completeley away. 

This time, though, as suggested, the figures mentioned have been revised accordingly.

Thank you again for helping us with your reviews.